# Coming-of-Age of Teenage Female Arab Gothic Fiction: A Feminist Semiotic Study

**Zoe Hurley** * and **Zeina Hojeij** *

College of Interdisciplinary Studies, Zayed University, Dubai 19282, United Arab Emirates
* Correspondence: zoe.hurley@zu.ac.ae (Z.H.); zeina.hojeij@zu.ac.ae (Z.H.)

**Abstract:** This feminist semiotic study explores the folkloric imaginary of the *jinn* in the context of children's and young adults' Arab Gothic literature. Across the Middle East, the *jinn* is a common trope in literature, folklore and oral storytelling who, in diegetic terms, can manifest as the Gothic figure of an aging female, deranged older woman or succubus (known as *sa'lawwa* in Arabic). In this study, a novel feminist semiotic framework is developed to explore the extent to which the Gothic female succubus either haunts or liberates Arab girls' coming-of-age fictions. This issue is addressed via a feminist semiotic reading of the narratives of Middle Eastern woman author @Ranoy7, exploring the appeal of her scary stories presented on YouTube. Findings reveal tacit fears, ambivalences and tensions embodied within the Arab Gothic sign of the aging female succubus or *jinn*. Overall, the research develops feminist insights into the semiotic motif of the female *jinn* and its role in constituting Arab females as misogynistic gendered sign objects in the context of the social media story explored.

**Keywords:** Arab children; literary imaginary; feminist semiotics; female-homosocial spaces; teenage fiction; Gothic; coming-of-age; social media; fictions





## 1. Teenage Female Arab Gothic *Bildungsroman*

Within the patriarchal norms of Middle Eastern societies, the Gothic offers a theoretical point of entry to consider *bildungsroman* narratives depicted within Arab teenage and children's literature. *Bildungsroman* is a literary genre that explores the psychological development and coming-of-age of protagonists from childhood to adulthood (Lynch 1999). This feminist semiotic study addresses the lacuna in scholarship concerning Arab children's and teenage literature on YouTube and explores how the Arab Gothic trope of the *jinn* renders the aging female as irreal, fantastical and monstrous. Within Middle Eastern culture, the figure of the *jinn* refers to a group of beings living alongside humankind who can traditionally be either male or female. They are believed to have the capacity for both good and evil and, while different from humans, they are defined as "a form of elemental life, psychic and invisible but not purely spirit" (Olomi 2021, p. 145). This study considers why *jinn* frequently manifest as Gothic female figures in Arab young adult and teenage fiction. According to Billone (2016), interest in the Gothic has escalated in the 21st century and signals a preoccupation with "the end of innocence". In the case of the Arab Gothic, *jinn* represent an anxious *bildungsroman* in the form of "deranged" older women, "hags" in disguise or *sa'lawwa* (a succubus—a demonic female creature). The central question of the study is how can a feminist semiotic reading help to theorise the gendered sign of the *jinn* in female Arab teenagers' popular imaginaries?

To address these issues, the study develops the framework of feminist semiotics (Hurley 2021a). Semiotics is concerned with signs (defined as symbols, letters, words, narratives, images, videos or any phenomena conveying meaning). According to the semiotician Leone (2013, p. 115), drawing on Yuri Lotman's (1990) cultural semiotics, subjectivity can be understood not only as physical, philosophical or psychological phenomena but also,

and perhaps most significantly, as a semiotic construct that societies and cultures shape through complex accumulations of signs. In this paper, informed by the cultural semiotic perspective, we argue that gender is also a profoundly semiotic entity that culminates via a series of signs, which are physical, psychological, conceptual and hierarchical. Feminist semiotics is developed as an expansive lens. It is designed to systematically explore the semiotic motifs of the *jinn* as sign vehicles for constituting a hierarchy of gendered identities. In what follows, we focus our study on the *jinn* narrative of the young woman YouTuber @Ranoy7 from the United Arab Emirates (UAE), who tells "scary stories" or Gothic urban legends in Arabic to her 1.65 million young followers (Hurley 2023). This study is important because it contextualises semiotic meanings while theorising the tacit ideological significance of the *jinn* within contemporary Arabic young adult and teenage fiction. To consider these matters, we begin with a review of relevant academic literature and outline the feminist semiotics framework and methodology and then go on to present the findings, followed by the discussion and conclusions, of the study.

## 2. Reading the Teenage Female Arab Gothic

To carry out this literature review, we conducted a thematic review of relevant scholarship addressing Arab children's Gothic literature. Using the Boolean search terms female/Arab/Gothic/children's literature/teenage fiction/social media storytelling/jinn, we explored academic databases, scholarly texts and articles. These themes reveal that, while the Arab Gothic represents a gap in scholarship, the Gothic is a popular genre in Middle Eastern fiction.

The concept of *jinn* is rooted in Arab culture and is an integral part of pre-Islamic and Islamic storytelling (Ibn Kathir 2020, d.1373). The *jinn* are mentioned in the Qur'an several times and have an entire *sura* traditionally named after them. In the 72nd chapter—*Sūrat al-Jinn*—the Qur'an explains the relationship of the *jinn* with God and the Prophet Muhammad (may peace be upon him) (Olomi 2021). *Jinn* also appear in the famous *1001 Nights,* which was Persian in origin but also translated into Arabic. The *Arabian Nights,* which is the English title for the collection of Middle Eastern folktales, was translated into English during 1706–1721. The stories were collected over several centuries and by various authors, translators and scholars across West, Central and South Asia and North Africa. Some tales trace their roots to ancient and medieval Arabic, Persian, Indian, Greek, Jewish and Turkish folklore and literature while moving over multiple locales, including the great empires of China, India, Turkey and Persia (Hurley 2021b). A common trope in all the editions of the *Nights* is the imaginary of young women's sexual vulnerability. In numerous versions and translations of the *Arabian Nights*, a sub-set of stories are told by the young female Scheherazade, who describes fantastical horrors facing Arab girls and women. Within the main diegesis, Scheherazade, to stay alive, tells her sister these narratives from her bed chamber while captivating the eavesdropping despotic King Shahrayar to prevent him from killing her, as he has done to other virgins before her.

Beyond the narrative realm, the Middle East and North Africa (MENA) region has been plagued by colonialism, wars and crises while undergoing rapid periods of urbanization, economic expansion, political shifts and displacement of many communities. Although there are significant differences between the territories, languages, religions and cultures of the MENA, beliefs in the *jinn* stretch throughout the region. Several MENA communities share a belief in *jinn* regardless of their urban, suburban or rural background. Education and level of devotion, rather than location, play roles in these beliefs, which are a widespread—albeit controversial—aspect of popularist Arab Gothic imaginaries. According to Fred Botting (1996, p. 3), Gothic interventions typically emerge when "imagination and emotional effects exceed reason". Simultaneously, since teenage girls' thoughts, desires and traumas are not overtly articulated in contemporary MENA societies, the Gothic has become a fertile literary genre for children, teenagers and young adults, especially on the Internet. In particular, the Gothic tropes of the female *jinn*, in the Arabian Gulf and Levant, reveal acute anxieties about the horrors of girls' and women's aging. However, the extent

to which the Arab Gothic might be considered a feminist discourse, which is a claim that Douglas (2021) makes for non-diasporic Arab women's Gothic literature, is ambiguous. We discuss these ambivalences further in the following section.

Shared features of teenage female Arab Gothic (TFAG) fiction, manifesting in the figure of the *jinn*, are apparent across the MENA and evident in the reoccurring motif of "monstrous" aging girls. To understand the "horror" surrounding the aging Arab female, who is pre- or post-pubescent or has long come of age and is menopausal, we note that the MENA gendered context is not defined by an absence of a female public sphere but rather by homosocial spaces of women and girls, who are often positioned to observe each other and compete and jostle against one another, at weddings, beauty salons, female-only malls, schools and universities and on social media (Le-Renard 2014; Hurley 2021b). For instance, in the Gothic folktale "*Jubaybani*" (Nuweihed [2007] 1907), a young princess is protected "against every trifling breeze, so that she grew up without so much as passing beyond the walls of the garden around her, seeing nothing of the outside world outside her palace" (p. 63). Due to this feminine confinement, the young Jubaybani " . . . is still unsatisfied, and wished to wander further in those open meadows, to further meadows still" (p. 65). In another story translated by Nuweihed (Nuweihed [2007] 1907), the female sphere (despite its "benefits" and "garden, the servants, delicious food and comfortable sleep" (p. 205)) is despised by the male protagonist, who refuses to be kept as a "prisoner" since, as he states, "I'm a free man, who goes wherever he pleases and sleeps wherever he wishes" (p. 205).

Unlike female characters, male protagonists across Arab folklore, cinema and musical genres are defined by the freedom and adventures of the street. The politicised drama of the street is a central theme in Egyptian narratives and the genre of "*shaabi*" (Hamam 2013, p. 186). *Shaabi* means "of the people". It originated in Cairo in the 1970s as a new form of urban music expressing the difficulties and frustrations of modern Egyptian life. *Shaabi* lyrics can be intensely political yet filled with humour and double entendres (Hamam 2013). Conversely, while *shaabi* is an amusing trajectory for Arab male protagonists, for the young Arab girls who have been traditionally closeted away, the feminine sphere is a profoundly serious place. Within the context of the Arab Gothic, claustrophobic female-only spaces are also a source of dread within teenage female Arab Gothic narratives.

To provide further background to these Gothic imaginaries, we can consider Elabed's (2016) point that emotional abuse, verbal bullying and derision of women by their female elders is prevalent in Middle Eastern families, in which "girls are viewed merely as daughters to be married, never permanent in their family's home and daughters-in-law are outsiders, squatters among their son's or decorated maids and baby-machines". For instance, *The Story of Zahra* (Al-Shaykh 1986), *Maryam's Maze* (Ez-Eldin 2007) and *The Seamstress' Daughter* (Haddad 2019) are Gothic tales of depressed mothers' and girls' descents into madness in a patriarchal world that denies Arab female ontologies (Douglas 2021). The Arab female sphere is thus often marked, in folklore, children's literature, adult fiction, film and lived experience, by the vulnerabilities of women, who are placed not only at the mercy of the patriarchy but also as misogynistic objects, including the unwanted first wife, vengeful mistress, mother-in-law, abused maid, widow, stepdaughter and second-class citizen. Confined within the narrow roles of the Arab female sphere, as Middle Eastern females age, they act out their frustrations. Older women can emerge as an object of "horror", while their aging caucuses, peri- and post-menopausal bodies, wrinkled skin and unkempt hair are terrifying signs that women's and girls' bargaining power within patriarchy depends on their role as potential mates, child bearers and objects of attractiveness.

Beyond Arabic fiction, the character of the older, "grotesque" woman (who can no longer conceive) is a common trope in fairy tales, ranging from the "evil" queen in *Snow White* to the "wicked witch" in *Hansel and Gretel* or the *Baba Yaga* in Slavic folklore. This trope echoes the Gothic narratives of 19th century English fiction; for instance, the *bildungsroman* of the vulnerable female orphan and eventual school governess from Charlotte Bronte's *Jane Eyre* (Bronte 2019). In Bronte's *Jane Eyre*, we also meet the character of the intimidating Bertha Mason, who is locked away by her husband Mr. Rochester in the attic of Thornfield

Hall. She is an ominous figure, full of uncontrollable passion, violence, sensuality and madness, almost bestial in her behaviour. Bronte writes:

> In the deep shade, at the farther end of the room, a figure ran backwards and forwards. What it was, whether beast or human being, one could not, at first sight tell: it grovelled, seemingly on all fours; it snatched and growled like some strange wild animal: but it was covered with clothing, and a quantity of dark, grizzled hair, wild as a mane, hid its head and face. (p. 70)

The disturbing ambiguity of this scene stems from the incongruity of the figure's species, gender and physical precariousness. It defies human etiquette and is devoid of personal grooming, comprehensible language or any respect for the conventions of bourgeois gendered space. While Bertha could be considered as a binary for Jane's controlled nature and conventional appearance, the wild woman is a mirror vision of Jane's repressed passions and the rebelliousness she displayed as a child. As in the story of *Jane Eyre*, Billone (2016) suggests that all children's literature can be considered *bildungsroman*, while the global Gothic continues as a central trope of our technology-driven age, a form of sleep deprivation and/or nightmare endemic to late capitalism (p. 2). Just as Carter (1974) said, "we live in Gothic times", we believe that the Middle East is defined by its indigenous Gothic features, political repression and the fantastical authenticities of its social media (Hurley 2019). However, whether these stories and the TFAG tales circulating on the Internet characterise a feminist imaginary is yet to be seen and is a central issue of this article. Next, we discuss the novel feminist semiotic methodology of the study employed to further explore these matters.

## 3. Feminist Semiotics Framework

A feminist semiotics framework involves the study of signs, including words, images, audio, imaginaries and any phenomena conveying meaning (Hurley 2021a). Identifying key features of semiotic systemic coherence—or notable signs and structures and fluid sign interpretations—allows the semiotician, via systematic cultural analysis, to single out and define, at least hypothetically, the texts of a culture (Leone 2013, p. 118). However, as feminist semioticians, we are aware that semiotics' perennial problem could be the binary of researcher and researched (Hurley 2021a). Indeed, Leone (2013, p. 117) admits that, "Cultural semioticians skate on thin ice". For instance, different semiotic cultures' fictional histories can be difficult to map in comparative studies due to varying languages, ontologies and epistemologies. There is also a growing body of literature indicating that translation of children's literature between Arabic and English or vice versa can be problematic, and direct equivalency between texts is unlikely (Dillon et al. 2018; Hojeij et al. 2019). Translation is also a subject of interest to semioticians, who consider meaning as occurring through the semiotic translation and combination of signs (Welby 1911; Petrilli 2014).

However, it can be challenging to translate sensitive semiotic cultural subject matter, such as the gendered features of teenage girls' lives, which are not overtly represented in Arab culture (Dukmak 2012). In the case of English-to-Arabic and/or Arabic-to-English translation, the challenges stem from attempts to not only bridge together contrasting languages and orthographic scripts but to connect varying cultures and semiotic norms surrounding children's gendered lives and identities. What tends to be translated, in the case of English-to-Arabic and Arabic-to-English translation, in children's literature comes from cultural interpretation, additions and frequent flattening of the original (Hojeij et al. 2019). For instance, English translations of the Arab folk stories by Khoury (2013) in *Pearls on a Branch* emphasise the trajectories of female protagonists. For example, the story *The Girl Who Had No Name* (pp. 148–59) is about a princess who, despite being "hidden in an underground vault and there raised until she grew up", still manages to learn to read and write and to charm a prince into marrying her. This emphasis on the protagonist's literacy is a detail that might make the story more palatable for contemporary readers, but it is, nevertheless, a reinterpretation of the original. However, despite evidence of these intercultural translations, there is an evident gap in research on teenage female

Arabic Gothic fiction, as well as a lack of theorisation of the semiotic tropes, tensions and ambiguities of its reinterpreted content and appeal.

From a feminist semiotic perspective, a further thorny issue in semiotic cultural analysis arises when semioticians assume that they can stand back from empirical experience and use theory to analyse subjects as objects. From a feminist perspective, such a framework is difficult to rationalise if it does not engage with the subjects or include women's polysemic voices in the analysis (Hurley 2021a). Thus, there is a danger that a detached version of semiotics is not aligned with feminist scholarship, which primarily seeks to give voice to women on the margins of power (Godard 2003). Nevertheless, Russian semiotician Semetsky (2017), while drawing on psychoanalytic feminism, emphasises the potential of feminist semiotics. According to Semetsky, semiotics (similarly to psychoanalysis) provides a tool to consider what cannot be easily articulated or is repressed; for instance, the uncanny meanings of signs surrounding gender. The novel feminist semiotic framework of this enquiry, building on semiotics, culminates as a method that can address these limitations by enabling the tacit meanings of the semiotic to be explored at the following three levels:

(1) Listing of the observable sign elements of TFAG fiction through a case study of @Ranoy7's social media storytelling;
(2) Analysis of the Gothic objects of meaning in @Ranoy7's social media storytelling;
(3) Consideration of the above to develop a feminist semiotic interpretation of the gendered significance and ambivalences of TFAG imaginaries.

First, developing an interpretive analysis of the TFAG case study involved Arabic-to-English translation and textual transcription of a series of @Ranoy7's YouTube videos. This was carried out in conjunction with three first-language Arabic-speaking research assistants. Second, several face-to-face interviews with the YouTuber @Ranoy7 were conducted to explore her perspective on the content she is creating. Third, feminist semiotic analyses of all the above were undertaken in combination to further develop the interpretation. The corpus of @Ranoy7's social media content was stored in the qualitative software portfolio Pathbrite, which enabled videos, images and monologues to be translated from Arabic to English and depicted within a single dashboard. An overview of this corpus is given in Table 1.

**Table 1.** Corpus of @Ranoy7 YouTube videos.

| Date | Title—English | Title—Arabic | YouTube Link |
|---|---|---|---|
| 17 July 2021 | *The Secret Behind the Girl in the Wardrobe* | السر وراء فتاة الخزانة | https://www.youtube.com/watch?v=XkDvJZWIHbM |
| 31 July 2021 | *Haunted Places Series \| Island of the Black Death (Bubonic Plague)* | جزيرة الموت الأسود | https://www.youtube.com/watch?v=06mrWZo5brA |
| 3 July 2021 | *Don't Peek Behind the Door: The Sa'lawwa (succubus)* | السعلوه الا تنظر خلف الباب | https://www.youtube.com/watch?v=KTf79pJDmKI&t=419s |
| 28 May 2019 | *You Won't Believe the Terrifying Truth Behind Disney Movies \| 4 Terrifying Fairy tales by the Brothers Grimm* | 4 الان تصدق الحقيقة المرعبة وراء أفلام ديزني حكايات خرافية مرعبة بواسطة الأخوين جريم | https://www.youtube.com/watch?v=Z4JHVtmspqM |
| 8 May 2019 | *Horror Stories Inspired by the Myths of Asia* | قصص رعب مستوحاة من أساطير آسيا | https://www.youtube.com/watch?v=bqLbrDAFKxU |
| 24 January 2019 | *Horror Story that Occurred in Egypt—Ghosts Living in a Family Home* | قصة رعب حدثت في مصر.. أشباح تسكن في منزل عائلي | https://youtu.be/1UlzzgVrAkI |
| 25 July 2019 | *Series of Mythical Creatures* | أم الدويس اسلسلة مخلوقات أسطورية | https://www.youtube.com/watch?v=6LJNVIP2ZCM |

Following the construction of @Ranoy7's storytelling corpus, each of the videos was analysed according to the analytic nodes listed above. This helped to address the central research question of the enquiry: how can a feminist semiotic perspective help to theorise the gendered sign of the *jinn* in female Arab teenagers' popular imaginaries? While

addressing this question, ethical issues were taken seriously in this study, adhering to Ackerly and True's (2008) belief that feminist research should be deeply committed to improving the quality (the knowledge claims) of women's scholarship. We obtained fully informed consent from the YouTuber @Ranoy7 to include her content in our analysis and gave her the opportunity to withdraw from the study at any stage without penalty. As feminist researchers ourselves, self-reflexivity is a further important component of our analysis.[1] In the following section, we move onto a presentation of our findings and a theoretical discussion of @Ranoy7's scary stories to consider these issues.

## 4. Observable Signs

### 4.1. Observable Elements of TFAG Social Media Stories

As indicated, @Ranoy7 is a popular social media storyteller with 1.6 million followers on YouTube, 96,000 on TikTok and 364,000 on Instagram. In her YouTube videos, her monologues are edited with the flickering effects, crackling screens and montages of Gothic images (see, for instance, Figure 1, providing a collection of screen grabs from @Ranoy7's YouTube narrative).

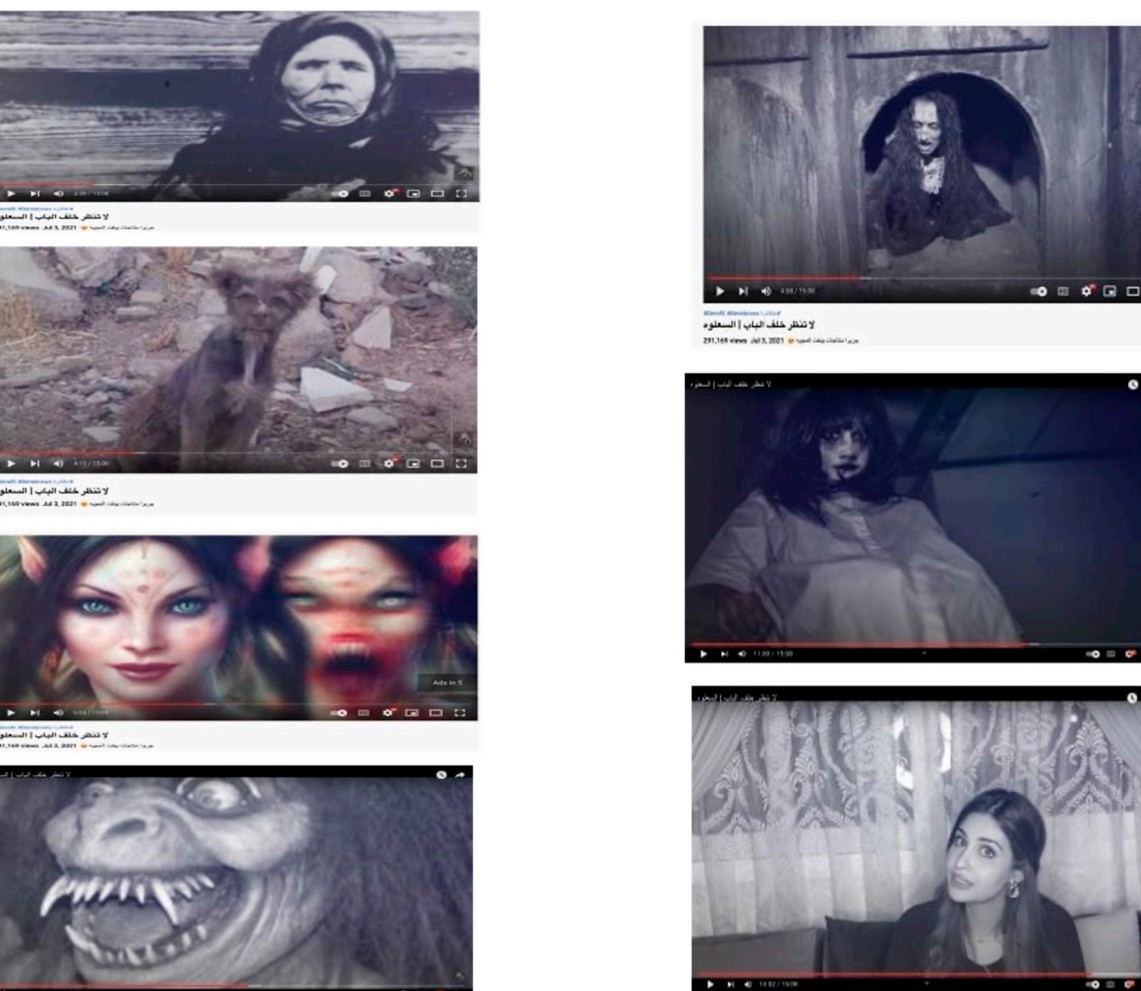

**Figure 1.** Screengrabbed with permission from @Ranoy7's (2021) episode *Don't Peek Behind the Door: The* Sa'lawwa (succubus).

Figure 1 shows screengrabs of the range of female characters included during @Ranoy7's narrative. Apart from the last image of the author, @Ranoy7 informed us that these are not original images and that they were selected from pictures circulating on the internet. These images include an aging veiled woman, a woman/dog, a female succubus with

pointed ears, a succubus with pointed fangs, another succubus, a demon girl and an image of the storyteller herself—@Ranoy7. The aesthetic illustrated in Figure 1 is a nostalgic and contrived sensibility reminiscent of the 1999 horror film *The Blair Witch Project.* This includes black and white documentary-style home-made video, known as found footage, that has seemingly been shot with a hand-held shaky camera and is used to bear witness to a series of unfortunate events. The found-footage genre is an international film cycle that has been growing in traction since the 1990s, the genesis of which can be traced to the Italian *Cannibal Holocaust* (Deodato 1980), which displayed mock found footage of the tragic deaths of a TV crew shooting a film in the Amazon within the context of a fictional narrative. Film theorist Sayad (2016) informs us that found-footage films are not necessarily presented to us as "inspired by" real events, but they constitute the audiovisual documentation of the uncanny. What we see, it is implied, are real people and mythical creatures, not characters based on them. This uncertain combination of fictional elements, low production values and recycled semiotic images collapses the boundaries separating the narrative diegesis from reality. There is also the implication that the fictional realm could spill over into the lived context.

Throughout her YouTube channel content, @Ranoy7 tells us that she draws on the amateur aesthetic of lost-footage horror to signal the unreliability of technology, as well as the fantastical authenticity of female knowledge, folklore and post-truth stories. In the episode *Don't Peek Behind the Door: The* Sa'lawwa (succubus) (Figure 1), @Ranoy7 (shown in the last frame of Figure 1) addresses her predominantly teenage female audience directly and fosters close parasocial relations typical of social media influencers (Hurley 2019). Following these subtle parasocial formalities, as calibrations of engagement, @Ranoy7 promotes a series of products and brands to her young female audience and encourages them to buy products while consuming her narratives. The following excerpt from her video story, which streamed on 3 July 2021 (Figure 1), begins without audio. This helps to focus attention on @Ranoy7's face as she applies the makeup brand of the paying sponsors of the scary story episode. @Ranoy7 tells her teenage audience in Arabic:

> Hello my loves! How are you? You're all gracing my channel. Of course there's a new video today but before starting it, I would love to tell you about and thank today's sponsor "Benefit Cosmetics". You don't know how much I love this brand, most of my makeup that I use is from Benefit. Seriously, like, everything—mascara from Benefit, eyeliner from Benefit, eyebrow makeup from Benefit, and sometimes I use their tint which is so cute...If you haven't watched the channel before, [it] is about creatures whose stories are legends, so not real, but there are some cultures/societies—Arab and foreign—who believed these legends and believed they were real. So are they real or not? We don't know, but they're folklore passed down from ancestors to their descendants and so on until they became well-known in most countries.

The promoted content of her post is seamlessly inserted via parasocial engagement with audiences, who are directly interpellated as "Arab". @Ranoy7 speaks to viewers in first-person Arabic (translated into English for the purpose of this study) and as if they are having a two-way conversation. From here, she moves into the narrative diegesis of her video:

> Title: The sa'lawwa (succubus): Our story for today is the story of the sa'lawwa (succubus). The succubus is a demonic creature, a female, and she always disguises herself as a beautiful woman. Remind you of someone? Wait. She always disguises herself as a beautiful woman, but in reality, she is an extremely ugly creature with thick hair and looks like a monkey. Her legs are like a donkey or goat's. So her visage is very ugly, and she also looks very old.

In this passage, describing the beast-like state and excessive body hair of the succubus, we are reminded of Bronte's Bertha who was referenced above. However, @Ranoy7 is more explicit about the "demonic creature['s]" unbridled appetite and semiotic disguises:

But when she sees a man that she may want/desire, or sees a young child she wants to lure, what does she do? She turns into this gorgeous woman, tall, blonde, and then seduces men or lures children who follow her as she takes them into her lair where she kills and eats them. In other stories, they used to say she kidnaps men on rivers, keeps them for years and marries them, and has children with them—and after years, returns the men to their families. These are some of the things that were said about the succubus [in stories]. The term "sa'lawwa" is used for a tall, skinny, and ugly girl. So these three characteristics all at once, I don't know how to be honest. That's the reason they called this creature a sa'lawwa. This creature-lore is still passed on in Iraq and the Arabian Peninsula and even in the Jordanian Badia (steppe). This creature is discussed and known within the residents there as a horrifying, nocturnal kidnapper of men and children and women. Every country or city says that the creature kidnaps a certain person, but they all agreed that she is very ugly, very hideous, and scary.

In this description, @Ranoy7 links the Arabian geography of Iraq, the Arabia Peninsula and Jordan to construct a shared community between her Arab viewers and to establish a sense of place. Within this geopolitical imaginary, she suggests that the danger in the region comes from "a horrifying, nocturnal kidnapper of men". In other words, horror stems from the imagined agency of female *jinn* who might identify "a man that she may want/desire". The succubus then "kidnaps men on rivers, keeps them for years and marries them, and has children with them—and after years, returns the men to their families". According to @Ranoy7, this mode of female dominance is "very ugly, very hideous, and scary". To understand the tacit ideologies of this narrative, in the next section, we turn to the analysis of the objects of meaning of @Ranoy7's monologue.

### 4.2. Objects of Meaning

Content analysis and interviews with @Ranoy7 revealed that her objective is to help her teenage and young adult female followers engage with Arab stories and (pseudo-) historical events. In doing so, she is also interpreting some of the contradictions of the Arab gendered sphere. @Ranoy7 is thus not only telling her young female audience scary stories but also simultaneously analysing their significance as objects of meaning. Following on from the above monologue, she tells her viewers:

There are some families around the Euphrates River in Syria who think that she is a demonic creature, a devil—that she is a jinn who shows up when dark/black magic is performed in a grave, where an old woman enters a grave and comes out as a dog, enters another grave and comes out as a crow . . . So her story is very similar to many stories traded in Arab countries. The story or the creature is always similar in characteristics but the name is different. So why did these people in previous times, though there wasn't Internet or social media platforms, there wasn't anything that could spread these stories so much—so how did each country—like, where is Egypt and where is the UAE, you know? The distance! How then did the Naddaha legend which is the same as Umm Al- Duwais become well-known in both countries? Why with the same characteristics, same thing, and kidnaps men? I suspect that women were making up these stories so that their husbands don't look at/seek out beautiful women [laughs]. "How do I make sure my husband doesn't cheat? I'll tell him stories about a beautiful demon that comes out at night and kidnaps men who flirt with her? Makes sense."

The above illustrates that @Ranoy7's objective is to inform her audience about how Gothic stories can serve as a conceptual tool in the Arab patriarchal bargain or as means by which women can navigate contemporary gender inequalities. However, rather than making these discussions overtly political, the message is diffused via coding of imaginary Arab folklore. Literary scholar Douglas (2021) explains that Arab feminist Gothic writers also use the *jahiliyyah* period, which is considered to the "time of ignorance" before Islam,

as a folkloric referential imaginary to evoke the female condition under contemporary patriarchy. According to Douglas (2021), the presence of *qarina* spirit-doubles or *jinn,* although little acknowledged, can be considered as a characteristic of the MENA feminist Gothic mode. @Ranoy7, stepping outside of the narrative diegesis, explains to her viewers that folklore can help to ensure that *"my husband doesn't cheat"* by warning them of older, predatory and available women. She continues to build this Gothic female onto-epistemology of logos/pathos:

> I'll tell you a story that popularised the sa'lawwa, which was discussed at length in communities in previous times in order to frighten people, like children and husbands and everyone basically. The story's ending is lovely, but its beginning is scary. It starts with a man who has three beautiful daughters. This man, unfortunately his wife passed away, and he was alone in raising his three daughters. One day, he went to the market, and there he met a very beautiful woman—tall, long hair, beautiful—she was the perfect woman as they say. He chatted with her and liked her a lot. He went back home and the whole time only the woman was on his mind... He asked the woman for her hand in marriage, and she said yes but on one condition: That he and his girls would go to her hometown/city and live in her house. The father of course immediately agreed, having not believed she said yes in the first place, and they got married and relocated to her home.

Having established her authority as storyteller, @Ranoy7 is able to present a diegesis in which "everyone basically" should be wary of "very beautiful" yet assertive women.

> One day, after a while, the woman told her husband that the house didn't have anything left in it and told him to go to the market and purchase what was missing. The father agreed of course but told her that he doesn't know the area very well or where the market is, so she said I'll guide you to where it is. So the father followed her into a long, dark hallway which leads to the house door.

The father's movement through the succubus's "long, dark hallway" is an uncanny symbolic journey and, since it has been initiated by the *sa'lawwa,* the father is overpowered.

> He was walking and she was behind him so as to guide him, and the father was surprised by a room in the hallway he never saw/noticed before. As he reached the door of the room, he turned around to ask his wife what this room was and was surprised by an extremely hideous visage. A tall, hairy creature, with an old-looking face—this wasn't his wife at all. Her legs were that of a goat's and her teeth were large and jagged, and she was very, very ugly.

To understand the significance of the imagery above, we can consider that, in certain Middle Eastern cultures—for instance, those of the Arabian Gulf, Afghanistan and Egypt—some women are encouraged to cover their faces in public and to wear the facial veil or *niqab*. For instance, in *Memoirs from the Women's Prison*, Egyptian feminist activist El-Saadawi (1983, p. 29) describes the controversies surrounding Muslim women's facial presentations and observes that the "Two eyeholes" of a female inmate interrogate her. A disembodied voice asks, "Isn't a woman's face a blemish upon her, a shameful private part to be covered?" (cited in Hurley 2020, p. 647). El-Saadawi here indicates how her cellmate in the prison links the female face to a woman's private parts. This analogy could help us to consider how the succubus's "extremely hideous visage" might serves as a metaphorical metonym for the older woman's private parts in the story. @Ranoy7 tells us that the father character's venturing into the "room in the hallway" leads to the discovery of: "A tall, hairy creature, with an old-looking face—this wasn't his wife at all. Her legs were that of a goat's and her teeth were large and jagged, and she was very, very ugly". @Ranoy7 continues to describe how the father is overwhelmed by the female "visage" (which serves as metonym for a vagina dentata):

The father didn't even manage to speak before the succubus attacked him and bit him, dragging him into the room and killing him. She left the room and returned to her beautiful disguise, going back to the daughters.

The succubus's "beautiful disguise", although one of youth and innocence, masks her real form as a "very, very ugly" older woman. The narrative also reminds us that, when Arab girls are married, they leave the family home and the mirage of the safety of the patriarchal sphere is shattered. Even though this female predicament represents an abandonment of the daughter by her father, blame is attributed to aging female characters who are poor substitutes for the maternal mother. @Ranoy7 tells us:

After a while, since the father did not return, the eldest girl asked the woman where her father was as he still hadn't returned, voicing her concerns that he might have gotten lost or hurt, and asked the woman to guide her to where the market is. The succubus told her he surely is still shopping and to wait for tomorrow, if he doesn't come back then she would take the daughter to the market.

This passage indicates that the threat of older women is an anti-feminist trope that has been repeated throughout Arab and global folklore. While engaging with this story, a teenage girl's trepidation regarding abandonment by her father is displaced onto an imagined treachery of older women. It is thus symbolic of the betrayal by the maternal mother figure, who abandons her daughters in marriage not only to patriarchal husbands and fathers but also to mothers-in-law. @Ranoy7, rather than questioning these misogynist values, reminds audiences of the "horror" of the ill-disciplined female who does not remove her body hair or maintain the feminine signs of youth. Moreover, the succubus in @Ranoy's tale is all-consuming, hungry and insatiable.

And of course, the next day came and the girl saw that her father did not return, so she asked the succubus to tell her where the market is located. She said of course, come with me, come after me. And she took her to the same dark, scary hallway, and transformed and attacked the girl and dragged her also into the room. On the day that the succubus killed the eldest girl, she went back to the two younger girls and told them that their sister left for the market, but that there is a lot of meat today and if they'd like a mouth- watering meal. Of course, the poor girls accepted and she cooked them a yummy meal with delicious meat that they ate . . . Now, this succubus was feeding her the same meat every day until she grew older, a little chubbier from all the meat, and more beautiful.

Within the diegesis, the female homosocial space is a degenerate sphere of ungainly geriatric feminine rage, unsatisfied hunger and desire.

When she was older, the youngest girl told the succubus that she had to leave and look for her father and sisters, that it didn't make sense for them to be gone for all of these years. At this point in time, the succubus could not control her temper, as of course this girl spent all of the previous years, asking about her family, and she transformed in front of her without even taking her to the hallway. Having transformed into this large and ugly creature, the succubus screamed at her that she raised and filled her up on her father and sisters' flesh, and that she would eat her as she ate them.

@Ranoy7's monologue, therefore, arguably articulates an anti-feminist message in advocating that a lack of patriarchal boundaries will culminate in an ill-disciplined woman who is overweight, unattractive and voracious. Without knowledge of the marketplace and patriarchal order, the characters in this story descend into the unbounded appetites of senior women. @Ranoy continues:

Of course, the youngest girl was shocked at what she saw and what she was told, so she ran and reached the dark, narrow hallway and saw the room, entering it to hide. Inside there was the big shocker. She saw the remnants of her father and

sisters' broken corpses, cut off limbs strewn about, their clothes tattered . . . she realised that this was a succubus, who killed her family and made her eat them. She took her father's clothes, the dishdasha [men's white traditional Arab dress], and disguised herself in them and left the room.

"Broken corpses" and "cut off limbs" are indexes of castration, while the motif of the young female protagonist cross-dressing in her father's *dishdasha* (robes) helps to reinstate the sign hierarchy of masculinity as the pinnacle of patriarchy. While masculinity is presented as performative, dressing as a boy liberates the young girl in the story, who is finally able to access the male *shaari* (street). Disguised as male, she can escape the confines of female signs and the rancid realm of the succubus. @Ranoy informs her audience:

The succubus didn't think that the girl was going to hide in that room in the first place, so she didn't check there at the time. So the girl ran away from the house and walked for a while until she reached a marketplace. She was roaming there, and people of course thought she was a man because she was dressed like one, so she tried to find work, to find a job, to find someone to help her. There she found people who were working at the palace, who took her in and let her work as a servant at the palace.

Doubled as a girl-boy, or *boyah* in Arabic, she is rewarded with the affordances of patriarchal society:

She worked day after day as a servant in the castle but would go to her residence's roof at night and take off her masculine disguise and eat there. As the prince was walking by, he saw her on the roof without her disguise and discovered her secret, that she was actually a woman disguised as a male servant . . . And of course the prince and girl got married and lived happily ever after.

Having lost her father to the succubus, the young protagonist finally restores the sanctity of the patriarchal order via her own marriage and by finding a new male protector in the form of a husband. However, due to the uncanny pleasures of the Gothic, premised upon mirroring, othering and an unpredictable moral order, nothing is ever entirely as it seems. Patriarchal hegemony is under constant threat and order has to be retained at the stage of diegetic denouement:

This is how the story ended, but there is a funny part which is when another prince arrived in this city, who was this prince's friend. The prince asked his best friend how he met and married such a beautiful girl, and the prince told him how she was disguised as a man and was a servant in the castle. So, when the second prince went back to his castle, and saw one of his servants who had feminine features, he thought it might be the same case for him as well. So he also asked this servant to bring his breakfast for the next day, and asked him to remove his "disguise" as well. But of course this was a man and the second prince's hopes of such a similar romance story were not borne.

In this section of the narrative, gender itself emerges as a "disguise" and a performance. Further, the tacit meanings that emerge are possibly the repressed desire of the homosexual male, who can gaze upon himself rather than "very, very ugly" womanhood when his mate mirrors his own male form. In the end, the female subject is something that must be repressed—as object—for it to be made palatable within patriarchy. Beyond the diegesis, the need to cover up, make over and disguise the female "visage" is encouraged by @Ranoy7, who, as a YouTube content creator, relies on the brand sponsorship of cosmetic companies and the promoting of make-up and make-overs. @Ranoy7's final denouement, stepping outside of the narrative diegesis, is anti-climactic since she must ensure her audience's loyalty and return to her YouTube channel. She tells them:

So, this was the end of today's story! I hope this story was to your liking. Please don't forget to give the video a like if you enjoyed it and share it with your friends to let them watch the video, and support me in any way you can, like sharing

the video on your WhatsApp groups that you have, to your friends, on Snapchat, everywhere, mention the channel. Let our family expand. I love you so, so much my lovelies, don't forget to follow me on my social media: Instagram, Snapchat, and TikTok, the handles are in front of you on the screen. I will see you my lovelies, God willing, in the next video. Bye!

Similarly to the succubus figure herself, @Ranoy7 needs to retain her grip on the attention economy by promoting her own feminised self-brand and simulating parasocial engagement. With these points in mind, we next develop a feminist semiotic interpretation of @Ranoy7's content.

### 4.3. Feminist Semiotic Interpretation

First, in terms of observable semiotic elements, the YouTube story enables @Ranoy7's young female audience to learn about gendered sign hierarchies via the signs and semiotic filters of @Ranoy7's Arab Gothic narrative. @Ranoy7's story of the succubus constitutes a Gothic explanation and semiotic translation of the horrors of patriarchy and female signs. While being simultaneously thrilled and chastised by its dangers in a mode of horror and fascination, female audiences assume a non-binary gaze and identify with the affordances of *boyah* identities and gender as performance. As Cocks (2020, p. 5) suggests of Gothic fiction more generally, TFAG involves "antagonistic iteration of gender and form, one that both collapses differences in gender and secures its asymmetry". Considering the case of the TFAG narrative under discussion helps us to appreciate that, due to unresolved gender ambivalence, young women can make sense of their abandonment by their fathers while blaming their mothers, thus maintaining patriarchal order.

At the second level of analysis, concerning the object of meaning, feminist semiotic analysis reveals that the young teenage viewers of the story are instructed in the importance of keeping the female "visage" disciplined via a series of signs, including the veiling of unsightly hair and masking of any signs of defilement. TFAG coming of age is thus an impossible *bildungsroman* of feminine sign-maintenance, grooming, reconstructing and masking. However, there is no "happily ever after" within patriarchy, as the female *bildungsroman* displaces the horror of misogyny onto older woman. TFAG fiction thus plays a hegemonic role in didactically guiding young female audiences through—but never beyond—the darkness, ignorance and post-*jahiliyyah* of internalised female misogyny. Consequently, the extent to which TFAG can be considered as a feminist narrative (e.g., Douglas 2021) in this case is doubtful.

Third, at the level of situated feminist semiotic interpretation, @Ranoy7's YouTube channel is revealed as a theatre of postfeminism. Postfeminism is defined as discourse that suggests that women are already liberated and, therefore, no longer require or adhere to feminist epistemologies (Hurley 2021a). There are several feminist studies that consider how postfeminist discourses frame hegemonically attractive women and encourage use of the feminine via self-objectification (Banet-Weiser 2018). Simultaneously, postfeminism is a travelling phenomenon, and there is not a universal mode of feminisation across varying cultures. Arab women and girls, similarly to other social actors, must navigate a series of contradictory discourses entangled within gender hegemony, postcolonialism, Arab nationalism and intersectional gender histories. These navigations are not without challenges, contradictions and situated semiotic variances (Hurley 2021a). @Ranoy7's narratives thus offer literal and conceptual tutoring in Arab feminised individualism in which hegemonic Arab female beauty ideals are the solution. While preying on her young female audiences' deep-seated anxieties, @Ranoy7 is simultaneously influencing them to buy products, follow, subscribe and "like" her posts.

What emerges from the analysis are detailed insights into the contradictions of the postfeminist theatre of signs. This is not an emancipatory feminist stage, as Douglas (2021) might suggest, but conceptual lost footage of survival stories about the horrors of girlhood, technological surveillance and Arab female teenage terror of *bildungsroman*. Thus, rather than advancing a liberating feminist narrative, in this case, TFAG is a mode of

postfeminism and of what Lauren Berlant (2011, p. 32) calls "cruel optimism", whereby women and girls are schooled in hegemonic femininity. While presenting makeovers, hair removal and beauty tips for Arab girls, @Ranoy7 coaches her female audience to despise older women and the aging process, while conjuring Arab femininity through semiotic disguises, make-up and masks. Nevertheless, there are some fleeting moments of liberation within @Ranoy7's scary stories: small victories for the female succubus, *jinn* and cross-dressers. While gender is staged in performative terms via a series of masks, costumes and acts, TFAG frequently restores patriarchal order via violent denouements and stark warnings to not stray to the margins of the gendered semiosphere or to ever unveil the ungroomed, semiotically disguised female "visage". Building on these insights, we now offer some conclusions

## 5. Discussion: Masking Gender

The central question of the study is how can a feminist semiotic perspective help to theorise the gendered sign of the *jinn* in female Arab teenagers' popular imaginaries? The three-step analysis explores the tacit semiotic meanings of @Ranoy's YouTube story, revealing the aesthetic of found footage, indigenous folklore and Arab Gothic motifs surrounding the female *jinn*. Theorising suggests that, within TFAG onto-epistemologies—or ways of knowing and being—the *jinn* is constituted as a misogynistic object via layered multimodal sign-symbol indexes. In the case of @Ranoy7's scary YouTube story about the *jinn* succubus, we reveal a didactic tale of terror about the "visage" of aging woman, the unkempt geriatric female form and the insatiable appetites of older women. @Ranoy7, while promoting products and telling a Gothic story, takes on the role of a semiotic interpreter and postfeminist translator who helps to constitute the patriarchal bargain, masking, making over and refashioning female identity to appear more palatable, youthful and obedient. Simultaneously, her targeted content and mode of Arabic female parasocial address disrupt a monolithic or one-size-fits-all global or feminist Gothic style (Douglas 2021). In doing so, @Ranoy7 can also be thought of as a Scheherazade figure, whose stories promote feminised survival mechanisms while constructing cruelly optimistic narratives about postfeminism. Simultaneously, @Ranoy7 is herself an aging woman who is "helping" girls and young women come of age by learning to apply feminine disguises, much like a *jinn*, to avoid appearing older. While curating make-up masks, digital filters and hegemonic femininity, @Ranoy7 plays the role of postfeminist semiotic interpreter and coaches her female audiences in how to disguise the female aging process. Bearing these matters in mind, in the final section, we offer some concluding points.

## 6. Conclusions

Interpretation of the case of @Ranoy7's scary story reveals a dense constellation of semiotic data that embed gendered sign hierarchies and the constitution of patriarchal sign logics. Through this feminist semiotic reading, we can view the case as facilitating an isotopic or generalisable reading as a hypothesis (Leone 2013). Although this narrative is the stuff of fiction, and not representative of the diversity of gender discourses in the MENA, analysis reveals tacit, gendered sign meanings that have been built via the momentum of semiotic representations of female *jinn*. Travelling like tumbleweed, the sign of the female *jinn* reoccurs across centuries and geographies, among different social actors and the varying media and contemporary semiospheres of Arab consumerist culture. The novel feminist semiotic insights help us to consider the significance of @Ranoy7's scary stories as semiotic ideological vehicles embedding layered sign doxa about Arab girls, young women and aging females. These narratives, rather than necessarily being a liberating Arab feminist Gothic (Douglas 2021), are cultural schema embedding postfeminist semiotic instructions on how to mask, disguise and—quite literally—make-over any potential rebellious signs of Arab female-gender non-conformity. Analysis thus expands the knowledge claims of feminist semiotic theory (Ackerly and True 2008), since the theorising is also itself something of a theoretical feminist semiotic *bildungsroman*. We thus encourage other

scholars to explore the potential of feminist semiotics, carry out further semiotic studies of the MENA's popular cultures and speak to the MENA's young readers, audiences and content producers directly. Finally, despite the limitations of this case-specific analysis, this study applied a feminist semiotic reading to an example of TFAG while cracking the theoretical ice of this under-researched area.

**Author Contributions:** Conceptualization, Z.H. (Zoe Hurley) and Z.H. (Zeina Hojeij); methodology, Z.H. (Zoe Hurley); software, Z.H. (Zoe Hurley); validation, Z.H. (Zoe Hurley) and Z.H. (Zeina Hojeij); formal analysis, Z.H. (Zoe Hurley); investigation, Z.H. (Zoe Hurley); resources, Z.H. (Zoe Hurley); data curation, Z.H. (Zoe Hurley); writing—original draft preparation, Z.H. (Zoe Hurley); writing—review and editing, Z.H. (Zoe Hurley); Z.H. (Zeina Hojeij); visualization, Z.H. (Zoe Hurley); supervision, Z.H. (Zoe Hurley); project administration, Z.H. (Zoe Hurley); funding acquisition, N/A. All authors have read and agreed to the published version of the manuscript.

**Funding:** This research received no external funding.

**Institutional Review Board Statement:** Ethical review approval by Zayed University Ethics Committee.

**Informed Consent Statement:** Informed consent was obtained from all subjects involved in the study.

**Conflicts of Interest:** The authors declare no conflict of interest.

## Note

[1]    The first author, Zoe, is a woman academic, originally from the United Kingdom, who has spent her adult life in Malaysia, Kuwait, Brunei and the United Arab Emirates. The second author, Zeina, is a female academic who is originally from Lebanon, was educated in the United States and is currently located at a university in the United Arab Emirates. We are self-reflexive about our own positionalities as transnational scholars and acknowledge the diversity of women's lives in the MENA.

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
