# Peer review of "Coming-of-Age of Teenage Female Arab Gothic Fiction: A Feminist Semiotic Study"

_humanities, doi:10.3390/h12010019_

Round 1

Reviewer 1 Report

The authors’  paper ‘Coming-of-age of teenage female Arab Gothic fiction: A feminist semiotic study’ is truly  enjoyable to  read for not only its style of writing but  it is strong  link between  Arabic literature and  feminist  semiotic  that can fill the  gap in  literature in the  region.   This is a very strong article, very well written coupled  by sound  methodology  and  strong  analysis and discussion.

In fact, the idea/concept of  jinn is rooted  deep  in  Islamic/Arabic  culture.  Jinn has appeared in the  notoriously   famous  1001 nights (Persian origin but  also translated into Arabic) and  their scary  representation of many  forms and  function of jinn  was  an integral part  Pre-Islamic  storytelling ( Please, refer for the  references  below) which I think might be appreciated  by  Arabic  readers.  

  However, In Islam, Jinn have  portrayed  slightly  different.  For example  in Quran, Al-Jinn (Arabicالجن, “The Jinn”) is the 72nd chapter (sūrah) of the Quran with 28 verses (āyāt). The name as well as the topic of this chapter is jinn. Similar to angels, the jinn are beings invisible to the naked human eye. In the Quran, it is stated in that humans are created from the earth and jinn from smokeless fire. However, the Quran reduced the status of jinn from that of tutelary deities to that of minor spirits, usually paralleling humans.[   { وَمَا خَلَقْتُ الْجِنَّ وَالْإِنْسَ إِلَّا لِيَعْبُدُونِ } [الذاريات: 56Hence, Allah  has  created  Humans and Jinn for the  purpose of  worship.

And yet,  in Islamic/Arabic   folklore, jinn and  possessions of  humans  has been the  main interpretation used  in  such narrative, driving  human crazy, love-sick (preventing them  from getting married),  and evil (please, refer to the article below- Possession and Jinn).

A few   suggestion for the authors:

1) Jinn can be either  male or  female, it would be best if the authors can indicated they using a  female  jinn here and not possible a female  version of the jinn.

2) Although  1001 Arabian nights are mentioned, yet the link of  Jinn has not been introduced  nor linked with Arabic story-telling

3) Line 76: as he has done to 1,001 other virgins (Mazolph 2007).  This is  a wrong  fact.  It is called  1001  nights based on the number of nights not the number  of  virgins being killed by the order of the  King. So every night  counts  for every  story Shahrazad  narrates  her compelling  story preventing the King from murdering her in order to simply hear more.

4) Line 81: a belief in jinns as.  Correction Jinn is  also plural.

5) please, be consistent  with  reference  style throughout the paper when needed.

6)  Please, be  consistent with the  font  throughout  the paper.

https://journals.sagepub.com/doi/pdf/10.1177/014107680509800805

Fee, C.R.; Webb, Jeffrey B. (29 August 2016). American Myths, Legends, and Tall Tales: An encyclopedia of American folklore. ABC-CLIO. p. 527. ISBN 978-1-610-69568-8.

https://muse.jhu.edu/pub/27/article/486604/summary

 Ibn Kathir (d.1373). "Tafsir Ibn Kathir (English): Surah Al Jinn". Quran 4 U. Retrieved 10 April 2020

https://journals.sagepub.com/doi/pdf/10.1177/014107680509800805

Author Response

Dear Reviewer, 

Thank you for your detailed and constructive suggestions. I have revised the article and made the suggested revisions. This includes:

(1) noting that the jinn can be male or female.

(2) Providing a more detailed discussion of the origins of 1001 Nights.

(3) Discussing the portrayal of jinn in Islam.

We appreciate your time and careful analysis of this article.

Yours faithfully,

Zoe Hurley

Reviewer 2 Report

I inserted many comments and edits into the manuscript. Please see the attached file.

Author Response

Dear Reviewer, 

Thank you for your detailed and constructive suggestions. I have revised the article and made the suggested revisions. This includes:

(1) correcting the typos, issues with punctuation and grammar in both English and Arabic.

(2) Strengthening the postfeminist analysis of the article and making it clearer that the YouTuber's text is not a feminist narrative.

(3) Attending to the miscellaneous errors and suggestions made by the reviewer.

We appreciate your time and careful analysis of this article and believe that it has been improved as a result.

Yours faithfully,

Zoe Hurley